Responsiveness of domesticated goats towards various stressors following long-term cognitive test exposure

Rosenberger Katrina 1 2 katrina.rosenberger@agroscope.admin.ch
Simmler Michael 3
http://orcid.org/0000-0002-1170-5431 Langbein Jan 4
http://orcid.org/0000-0003-4582-4057 Nawroth Christian 4
http://orcid.org/0000-0002-3663-7682 Keil Nina 2 nina.keil@agroscope.admin.ch
1 Graduate School for Cellular and Biomedical Sciences, University of Bern , Bern , Switzerland
2 Centre for Proper Housing of Ruminants and Pigs, Swiss Federal Food Safety and Veterinary Office, Agroscope , Ettenhausen , Switzerland
3 Digital Production, Agroscope , Ettenhausen , Switzerland
4 Institute of Behavioural Physiology, Research Institute for Farm Animal Biology , Dummerstorf , Germany
Vonk Jennifer
Electronic publication date: 2022 Mar 29
Publication date: 2022
Volume: 10
Electronic Location ID: e12893
Received 2021 Sep 3; Accepted 2022 Jan 16
Copyright: © 2022 Rosenberger et al.
Copyright year: 2022
Copyright holder: Rosenberger et al.
License: This is an open access article distributed under the terms of the Creative Commons Attribution License, which permits unrestricted use, distribution, reproduction and adaptation in any medium and for any purpose provided that it is properly attributed. For attribution, the original author(s), title, publication source (PeerJ) and either DOI or URL of the article must be cited.
License URL: https://creativecommons.org/licenses/by/4.0/

Keywords: Animal cognition, Animal handling, Habituation, Selection for productivity, Stress

Funding: SNF 310030E-170537 DFG LA 1187/6-1 The funders had no role in study design, data collection and analysis, decision to publish, or preparation of the manuscript.

==============================
Current evidence suggests that frequent exposure to situations in which captive animals can solve cognitive tasks may have positive effects on stress responsiveness and thus on welfare. However, confounding factors often hamper the interpretation of study results. In this study, we used human-presented object-choice tests (in form of visual discrimination and reversal learning tests and a cognitive test battery), to assess the effect of long-term cognitive stimulation (44 sessions over 4–5 months) on behavioural and cardiac responses of female domestic goats in subsequent stress tests. To disentangle whether cognitive stimulation per se or the reward associated with the human–animal interaction required for testing was affecting the stress responsiveness, we conditioned three treatment groups: goats that were isolated for participation in human-presented cognitive tests and rewarded with food (‘Cognitive’, COG treatment), goats that were isolated as for the test exposure and rewarded with food by the experimenter without being administered the object-choice tests (‘Positive’, POS treatment), and goats that were isolated in the same test room but neither received a reward nor were administered the tests (‘Isolation’, ISO treatment). All treatment groups were subsequently tested in four stress tests: a novel arena test, a novel object test, a novel human test, and a weighing test in which goats had to enter and exit a scale cage. All treatment groups weretested at the same two research sites, each using two selection lines, namely dwarf goats, not selected for production traits, and dairy goats, selected for high productivity. Analysing the data with principal component analysis and linear mixed-effects models, we did not find evidence that cognitive testing per se (COG–POS contrast) reduces stress responsiveness of goats in subsequent stress tests. However, for dwarf goats but not for dairy goats, we found support for an effect of reward-associated human–animal interactions (POS–ISO contrast) at least for some stress test measures. Our results highlight the need to consider ontogenetic and genetic variation when assessing stress responsiveness and when interacting with goats.

Introduction

Many animals in zoos, laboratories or research facilities are frequently confronted with cognitive tests for a scientific purpose, but little research has investigated the effects of these tests on the welfare of test subjects. Initial results suggest that cognitive stimulation via enrichment devices has positive effects on activity budgets and social interactions in primates (Yamanashi & Hayashi, 2011; Whitehouse et al., 2013; Jacobson et al., 2019) and the potential to increase exploration and reduce fear in farmed animals (Puppe et al., 2007; Zebunke, Puppe & Langbein, 2013). Positive emotions through the engagement in a solvable task (Hagen & Broom, 2004; Langbein, Nürnberg & Manteuffel, 2004; Meehan & Mench, 2007; Puppe et al., 2007; Manteuffel, Langbein & Puppe, 2009) and the reinforcing effect of the successful completion of a task have been suggested as potential explanations for these positive effects on animal welfare (Jensen, 1963; Hughes & Duncan, 1988). Others suggest that welfare is increased by the increased control over the environment (Meehan & Mench, 2007; Langbein, Siebert & Nürnberg, 2009). Zebunke, Puppe & Langbein (2013), for example, used a call-feeding station incorporated in the home pen, which called pigs by an individual acoustic signal when it was their turn to be fed as cognitive enrichment. The authors found that these subjects were less stressed by isolation and more exploratory towards a novel object compared to their conspecifics without this cognitive enrichment. Zebunke, Puppe & Langbein (2013) concluded that the introduction of cognitive stimulation in the housing environment has the potential to reduce stress responsiveness in future situations. Whether cognitive stimulation via standardised testing designed for a scientific aim affects stress responsiveness has not been investigated.

Whereas many devices designed to cognitively enrich animals are presented in the group within the home pen, cognitive tests used to assess animals’ cognitive capacities often require animals to be isolated from their group of conspecifics (e.g., Ruby & Buchanan-Smith, 2015), handled by a human, and/or given food reinforcement by a human (Morton, Lee & Buchanan-Smith, 2013; Nawroth, Von Borell & Langbein, 2014). Besides cognitive stimulation per se, the presence of a familiar human and the positive association with food may also contribute to behavioural changes and may reduce fear responses towards novel humans and ease handling in future situations (Boissy & Bouissou, 1988; Boivin et al., 1992). In addition, separation from the group is stressful for social animals such as cows and goats and was found to increase vocalisations, heart rate and cortisol levels (Boissy & Le Neindre, 1997; Da Costa et al., 2004; Aschwanden et al., 2008; Siebert et al., 2011; Patt et al., 2013) and to alter behavioural responses towards novel environments and towards handling (Veissier & le Neindre, 1992; Grignard et al., 2000). To assess whether standardised cognitive testing per se is associated with changes in stress responsiveness, it is necessary to disentangle the effect of testing from effects of confounding factors such as the human contact and the isolation during testing.

Several stress tests have been applied in animals to study the behavioural responses towards stressors such as isolation, handling by humans, and the confrontation with novel stimuli to various extents (Forkman et al., 2007). The so-called ‘open-field test’ or ‘novel arena test’ has been used to assess behavioural reactions towards isolation and a novel environment in many species (Prut & Belzung, 2003; Graunke et al., 2013; Oesterwind et al., 2016; Neave et al., 2018). Henceforth, we will refer to this test as ‘novel arena test’. The ‘novel object test’ has been applied to assess behavioural reactivity towards novel stimuli and to investigate the motivation to explore (Sneddon, Braithwaite & Gentle, 2003; Finkemeier, Langbein & Puppe, 2018). Research found that goats show signs of higher arousal and stress in novel arena tests such as higher vocalization rates and elevated cortisol levels during the tests (Siebert et al., 2011; Oesterwind et al., 2016). The responsiveness towards a human has been measured using animal–human encounter tests with a standing or walking human (Lyons, Price & Moberg, 1988) and with a sitting human (Romeyer & Bouissou, 1992). In addition to reactivity assessment using established stress tests in different farm animal species, stress responses can also be assessed in routine handling procedures such as weighing. The weighing situation includes several potentially stressful components such as human–animal interactions, novelty, confinement, and separation from conspecifics (Hemsworth, 2003; Forkman et al., 2007).

Stress responses towards the same stimuli can vary greatly between individuals. This inter-individual variation is caused by an interplay of environmental and genetic factors (Dantzer & Morméde, 1983) which in turn affect animal personality (Stamps & Groothuis, 2010). Breeding for specific traits can intentionally or unintentionally lead to changes in behavioural and physiological stress responses in the selection line (Rauw et al., 1998). Selection for high productivity has been shown to reduce stress responsiveness during human–animal interactions (Schütz & Jensen, 2001; Lindqvist & Jensen, 2008; Campler, Jöngren & Jensen, 2009; Colpoys et al., 2014) and to affect the responsiveness towards isolation (Romeyer & Bouissou, 1992; Kilgour & Szantar-Coddington, 1995). These differences could be relevant when exposing animals to handling procedures or novel environments. To achieve greater general validity of research results, it is therefore advantageous to consider testing different selection lines of a species. Goats are a suitable model species to study the effect of selection for productivity on stress responsiveness because not all selection lines were bred with the aim to increase productivity. Dwarf goats are farmed in extensive livestock production systems in West Africa and are generally not professionally bred. They are also often kept in ‘petting zoos’ or by hobby breeders in Europe. In contrast, dairy goats have been selected for high milk yield and ease of handling during milking. Goats are also suitable to study the effects of cognitive testing on stress responsiveness because common stressors such as isolation from the group (Price & Thos, 1980; Carbonaro et al., 1992; Aschwanden et al., 2008; Siebert et al., 2011), novel environments and objects (Forkman et al., 2007) as well as the presence of a human and handling procedures (Lyons, Price & Moberg, 1988; Lyons, 1989; Forkman et al., 2007) were already investigated in this species. Additionally, a lot of literature exists on the highly developed cognitive capacities of goats such as visual discrimination and social learning from humans (Langbein, Siebert & Nürnberg, 2008; Nawroth, Baciadonna & McElligott, 2016), and the motivation of goats to engage in cognitive tasks was demonstrated in previous studies (Langbein, Siebert & Nürnberg, 2009; Rosenberger et al., 2020).

In this study, we investigated whether long-term cognitive stimulation using object-choice tasks, including discrimination and reversal learning tests and a cognitive test battery, affects goats’ stress responsiveness in subsequent potentially stressful situations. We conditioned three treatment groups: goats individually exposed to human-presented object-choice tests (in the form of visual discrimination and reversal learning tests and a cognitive test battery, ‘Cognitive’, COG treatment), goats that received rewards from the experimenter without being administered the object-choice tests and thus could form a positive association between human and food without being cognitively challenged (‘Positive’, POS treatment), and goats that were isolated but neither received a reward nor were administered the tests (‘Isolation’, ISO treatment). All treatment groups were subsequently tested in four stress tests: a novel arena test, a novel object test, a novel human test, and a weighing test. We hypothesised that if long-term cognitive test exposure per se had a positive impact on behaviour and cardiac activity in subsequent stress tests, COG goats would differ from POS goats. Specifically, we expected COG goats to show a lower responsiveness towards a novel object and a novel human and thus to show fewer responses indicative of stress such as lower vocalisation rates, less activity, and lower heart rate than POS goats. Furthermore, we hypothesised that if long-term experience with reward-associated human–animal interaction had a positive impact on behaviour and cardiac activity in subsequent stress tests, POS goats would differ from ISO goats. Thus, we expected POS goats to show a lower responsiveness towards a novel object and a novel human and to show fewer responses indicative of stress such as lower vocalisation rates, less activity, and lower heart rate than ISO goats. To increase variability of our sample, we tested two selection lines of goats (dwarf and dairy goats) at two sites under comparable conditions (Würbel, 2017; Voelkl et al., 2018, 2020).

Material, Animals, and Methods

Location, animals, and housing conditions

We conducted this study at two locations, namely at the Centre for Proper Housing of Ruminants and Pigs at Agroscope in Ettenhausen (ET), Switzerland, and at the Research Institute for Farm Animal Biology in Dummerstorf (DU), Germany. As previously described in Rosenberger et al. (2021) we used two selection lines of goats; dwarf goats, not selected for productivity traits, and dairy goats, selected for high milk yield. In total, we studied the responses of 61 non-lactating female Nigerian dwarf goats and 59 non-lactating female dairy goats. We used dwarf goats bred in DU. The only selection purpose in this population was to avoid inbreeding. The potential milk yield of Nigerian dwarf goats does generally not exceed 0.3 kg/day (Akinsoyinu, Mba & Olubajo, 1977). As it was common practice in DU, dwarf goat kids stayed with their dams for 6 weeks before they were weaned. Regarding dairy goats, we used three of the most common high-producing dairy breeds in Switzerland and Germany and their crossbreeds, namely Saanen (n = 15), Chamois Coloured (n = 12), Saanen × Chamois (n = 3), and Deutsche Edelziege (n = 29). These breeds have potential milk yields of up to 3 kg/day (Vacca et al., 2018). In accordance with common practice in the dairy goat industry, the dairy goat kids were separated from their dam shortly after birth and artificially raised, e.g., by automatic milk feeders.

At ET, we housed 30 dwarf goats and 30 dairy goats (15 Saanen, 12 Chamois Coloured, three Saanen × Chamois crossbreds; see Supplemental File T1_animals). The dwarf goats were born in DU and moved to ET in June 2017. The dairy goats were born on different Swiss farms and were moved to ET in June/July 2017. At DU, we housed 31 dwarf goats and 29 dairy goats (Deutsche Edelziege; see Supplemental File T1_animals). The dwarf goats were born in DU, with the exception of eight animals. These were bought from the Zoo Osnabrück and the Wildpark Lüneburger Heide, Germany, due to shortage of female animals in the facility’s own breeding stock. The dairy goats were born on a German farm and were moved to DU in July 2018.

At the age of 7–8 months, all goats were moved to pens of 9–11 goats each, corresponding to three groups of dairy goats and three groups of dwarf goats at both locations. The total area of each dwarf goat pen was 14 m2 (approximately 3.6 × 3.9 m), including a deep-bedded straw area of 11 m2 (approximately 2.8 × 3.9 m) and a 0.5-m-elevated feeding place (1.4 m2). The total area of each dairy goat pen was 17.7 m2 (approximately 3.9 × 4.55 m), including a deep-bedded straw area of 13.4 m2 (approximately 4.55 × 2.95 m) and a 0.65-m-elevated feeding place (1.82 m2). Hay was provided behind a feeding fence at the feeding place twice a day at around 8 AM and 4 PM in ET and at around 7 AM and 1 PM in DU. Each pen had one drinker and a lick block for mineral supply. Additional structures in the straw-bedded area included a wooden bench (for dairy: 2.4 m long, 0.6 m high, 0.62 m wide; for dwarf: 2.3 m long, 0.5 m high, 0.5 m wide) along the wall of the pen and a round wooden table (0.8 m high, 1.1 m in diameter) in the centre of the pen.

We performed all animal care and experimental procedures in accordance with the relevant legislative and regulatory requirements of the corresponding country and the ASAB/ABS Guidelines for the Use of Animals in Research (ASAB & ABS, 2018). The Cantonal Veterinary Office, Thurgau, Switzerland (Approval No. TG04/17–29343) and the Committee for Animal Use and Care of the Ministry of Agriculture, Environment, and Consumer Protection of the federal state of Mecklenburg-Vorpommern, Germany (Approval No. 7221.3-1.1-062/17) approved all procedures involving animal handling and treatment. This study did not have to establish humane endpoints, as we either used goats for subsequent studies or gave them to new homes.

Treatment groups

As previously described in Rosenberger et al. (2021) three goats from each of the 12 pens (each housing 9–11 goats) were pseudo-randomly assigned to one of the three treatment groups: COG (n = 36), POS (n = 36), and ISO (n = 36). Except one pen, all pens housed one to two extra goats not assigned to a treatment group to replace others in case of e.g., disease or injury. Since arrival at the respective research farm, human-animal interactions were limited to the usual husbandry procedures only. In DU, the administration of the treatments took place in a room in the same building where the goats were housed. In ET, the goats had to be moved to a different building.

In 44 test sessions distributed over a period of 4–5 months, goats from the COG group were exposed to cognitive tests in the form of object-choice tasks to assess their discrimination and reversal learning skills as well as their ability to use physical cues (five clues plus control condition) and human gestures (five gestures plus control condition) to locate a hidden reward in a cognitive test battery (see Supplementary Text and Table S1 for details regarding habituation to test environment and treatments). During these tests, COG goats received food rewards from the experimenter for correct responses. The POS group was not exposed to cognitive tests but received a similar number of rewards as individuals in the COG group (= median number of rewards received by COG group in the previous test session), provided by the experimenter in the test arena at random times but over a similar amount of time as the COG group (= median time taken by COG group to finish all trials in the previous test session). Contrasting COG versus POS allows investigating the effect of the cognitive testing per se, disentangled from the effects of the positive association with the human and the isolation from the group during testing. Individuals of the ISO treatment neither participated in cognitive tests nor received rewards by the experimenters in the test arena. Instead, they were isolated over a similar amount of time as the COG and the POS group in the same arena (= median time taken by COG group to finish all trials in the previous test session). Contrasting POS versus ISO allows investigating the effect of the positive association of the human with food, disentangled from the effect of isolation from the group during testing. To control for caloric intake, ISO animals received the same number of food rewards as POS and COG goats, scattered over the floor of the waiting room (to avoid positive association with the human) before they were isolated.

Stress tests

Two to three weeks after the administration of the treatments we measured the responsiveness of all goats towards various stressors in four tests: a novel arena test (NA), a novel object test (NO), a novel human test (NH), and a weighing test (WH) in a mobile scale cage. In ET, all goats were between 15 and 17 months old when the first stress test (NA) started (mean ± SD: dwarf goats 509 ± 0.9 days, dairy goats 468 ± 3.5 days). In DU, the goats were around 18 months old at test start (mean ± SD: dwarf goats 557 ± 3.4 days, dairy goats 540 ± 0 days). We completed all tests in the same order and within 6 weeks at both locations. The facilities where the NA, NO, and NH were conducted were familiar to the goats from administration of the treatments. The WH took place in front of the goats’ home pens. During all tests, acoustic and olfactory contact between the test subject and its peers was possible.

Novel arena test (NA)

The first test we conducted, the NA, was used to assess responsiveness towards isolation in a novel environment. We measured goats’ responsiveness in an arena (3 × 5 m) with opaque walls (2 m high), a grid drawn on the floor (with 12 segments), and a start box (1 × 1 × 1 m) attached on the outside but connected to the arena (Fig. 1). Each subject was placed in the start box for 20 s to standardise the beginning of each test. After 20 s, the start box was opened, and the animal was allowed to enter the arena. Immediately after the goat had entered the arena, the start box was closed, and the test subject stayed in the arena for 5 min. We recorded the duration of being inactive and the time staying in inner segments, the frequency of vocalisations and of changes of segments, as well as the heart rate (bpm). Each goat was tested once in the NA on one of two consecutive days on which, for a given selection line and site, all NA were performed.

Figure 1 Picture of the arena (3 × 5 m) used for the novel arena test (NA), novel object test (NO), and novel human test (NH). The arena was divided in 12 segments which were drawn on the floor.

The numbering shown was used to identify the segments when encoding the animals.

Novel object test (NO)

With the second test we applied, the NO, we assessed the responsiveness of goats towards a novel object. In the arena described above (Fig. 1), the same procedure as in the NA was applied, and a novel object (DU: green bucket, 30 cm high, 50 cm in diameter; ET: brown bucket, 40 cm high, 30 cm in diameter) was placed in Segment 11 of the arena. We recorded the duration of being inactive and the time staying in inner segments, the frequency of vocalisations, object contacts, and of segment changes, as well as the heart rate (bpm). Each goat was tested once in the NO on one of two consecutive days on which, for a given selection line and site, all NO were performed.

Novel human test (NH)

The responsiveness towards a human was measured using the third test, the NH. In this test, the goat was confronted with a stationary novel human (completely unknown to the goats and always wearing a lab coat) standing in the back (Segment 11) of the arena (Fig. 1) and looking at the wall above the entrance gate not making eye contact with the goat. All else followed the same procedure as in the NA. We recorded the duration of being inactive and the time staying in inner segments, the frequency of vocalisations, human contacts, and of segment changes, as well as the heart rate (bpm). Each goat was tested once in the NH on one of two consecutive days on which, for a given selection line and site, all NH were performed.

Weighing test (WH)

Finally, we scored goats’ responses to handling during weighing in a mobile scale cage (FX 21A, Agro Sigmer, Dietikon, Switzerland; Fig. 2). The goats had never been weighed before the test or seen the weighing scale. The experimenter first led the test subject onto the scale, where it stayed for 3 min starting as soon as the gate of the scale cage was closed. If the goat refused to walk onto the scale, the experimenter gently pushed it. After the 3-min period, the experimenter opened the gate on the opposite side of the scale cage, and the goat was allowed to walk out. If the goat refused to walk out, the experimenter gently encouraged it to leave the scale cage. Two experimenters were simultaneously scoring how easily the goat entered the scale cage (entering score), how it behaved on the scale (weighing score), and how easily it exited the scale cage (exiting score); they used a scoring system adapted from D’Eath et al. (2009; Table 1). During weighing, the experimenters positioned themselves on either side of the scale cage within 1 m distance. Testing of all goats took place on a single day (one trial per goat) and was done by the same two familiar experimenters at each location.

Figure 2 Picture of a dairy goat standing on the scale and wearing the harness used to record cardiac measures.

Table 1 Scoring system used for scoring responses to handling during weighing (adapted from D’Eath et al. (2009)).

Score	Description	
Entering the scale cage	
1	Goat is very difficult to move and tries to escape; hard pushing or lifting of legs by experimenter necessary	
2	Goat is difficult to move into the scale cage and tries to resist; some pushing by experimenter needed	
3	Goat walks into the scale cage with little encouragement by experimenter	
4	Goat voluntarily walks or runs forward into the scale cage	
Weighing on the scale	
1	Goat moves around a lot during weighing, with many escape attempts, rearing, and vocalising	
2	Goat mostly moves, makes several escape attempts, vocalises	
3	Goat moves around a bit during weighing, makes max. one escape attempt, vocalises	
4	Goat stands mostly still during weighing or tries to lie down, no escape attempts, little vocalisation	
Exiting the scale cage	
1	Goat resists and is very difficult to push out of the scale cage	
2	Goat moves out of the scale cage after some pushing by experimenter	
3	Goat (slowly) leaves on its own accord once the door is opened	
4	Goat quickly runs out of the scale cage, no hesitation	

Behavioural and cardiac measures

The behavioural and acoustic responses were videotaped with a camcorder (ET: Sony HDR-CX240E; DU: Panasonic HDC-SD60) and an external microphone (Table 2). Additionally, cardiac measures of the goats were recorded in each of the four test situations: in the NA, NO, and NH for 5 min starting when the goat entered the arena, in the WH for 3 min starting as soon as the goat entered the scale cage. The goats were equipped with an electrocardiogram (ECG) acquisition harness (BioHarness® system, MLE120X BioHarness Telemetry System, Zephyr Technology Corporation, Annapolis, MD, U.S.A.) which was fitted tightly around the chest behind the front legs of the animals (Fig. 2). Before each use, ECG gel was applied on the parts of the belt containing the electrodes. The electrodes were positioned on the left side of the chest, with one electrode placed close to the sternum and the other electrode over the right scapula. The ECG was recorded at 250 Hz by a logger integrated in the BioHarness® system and transmitted live to a laptop with the AcqKnowledge software (v.4.4, BIOPAC Systems, Inc., Goleta, CA, USA). To minimise stress linked to the novel device, the first test, the NA, was preceded by 1 day of habituation to wearing the harness.

Table 2 Definitions of behavioural and cardiac measures that were included in the principal component analyses for the novel arena test (NA), the novel object test (NO), the novel human test (NH), and the weighing test (WH).

Test	Measure	Type	Definition	
NA, NO, NH	Total time inactive	Duration (s)	Animal is standing still; legs are not moving	
NA, NO, NH	Vocalising	Frequency	Animal is vocalising with open or closed mouth	
NO	Object contact	Frequency	Animal’s snout touches or is within 5 cm of the object	
NH	Human contact	Frequency	Animal’s snout touches or is within 5 cm of the human	
NA, NO, NH	Change of segment	Frequency	Animal moves to another segment with at least both front legs (see Fig. 1)	
NA, NO, NH	Staying in inner segments	Duration (s)	Segments 5 and 8 (see Fig. 1)	
NA, NO, NH, WH	Heart rate	Beats per minute (bpm)	Baseline-subtracted heart rate during test	
WH	Weighing score	Scores 1–4*	Mean score (see Table 1) given by experimenters during weighing on scale, from high (= 1) to low (= 4) stress	
WH	Exiting score	Scores 1–4*	Mean score (see Table 1) given by experimenters during exiting of scale cage, from high (= 1) to low (= 4) stress	
WH	Entering score	Scores 1–4*	Mean score (see Table 1) given by experimenters during entering of scale cageTable-Foot, from high (= 1) to low (= 4) stress	
Note:

* Scoring system according to D’Eath et al. (2009).

For each goat, a baseline recording of cardiac activity was taken, either in its social group (DU and ET: NA, NO, NH, DU: WH) or individually in front of its home pen (ET: WH), shortly before it was taken to the test room. Cardiac activity for the baseline was measured for 10 min for the NA, NO, and NH and for 5 min for the WH. Immediately after the baseline measurement had been taken, functionality of the harness was assessed and, if needed, readjusted, and the goat was led to the starting box of the test room for the NA, NO, and NH or in front of the scale cage in case of the WH.

For behaviour coding, the videos (with audio) were analysed with the Observer XT software (v.13, Noldus Information Technology, The Netherlands) to determine frequency and duration of behavioural responses (Table 2). To assess inter-observer reliability, scores from the main scorer were validated with that of a second scorer, also blind to the study, who coded eight videos from the NA and eight videos from the NO. Cohen’s kappa indicated excellent agreement between coders across all coded behaviours (all κ > 0.99, p < 0.001, all rho > 0.99, p < 0.001). In the WH, the reliability between the scorings of the two experimenters measured by Spearman’s rank correlations was very high (entering score: rs = 0.75, p < 0.001, exiting score: rs = 0.79, p < 0.001, weighing score: rs = 0.82, p < 0.001), thus we merged the scores into one score for further analysis. The ECG was processed using EasieRR (Rasmussen, Rosenberger & Langbein, 2020). R peaks were automatically detected using the software’s peak prominence algorithm (peak prominence set at 0.05–0.07 depending on the ECG trace) and were reviewed visually for ectopic cardiac beats, missed beats, and outliers by three investigators. Because there was too much noise in the data to analyse the whole 3-min (WH) to 5-min (NA, NO, NH) recordings of the stress tests, we decided to only analyse selected time intervals of 20 s within certain time windows. For the baseline recordings, this time window started after 60 s to avoid measuring the effect of putting the harness on. For the recordings of the stress tests, the test time was split in half, and we analysed 1–3 time intervals of 20 s in the first half and 1–3 time intervals of 20 s in the second half of each test per animal. We selected time intervals in which the heart beats on the ECG trace were clearly visible and the signal-to-noise ratio was adequate. If artefacts in heart rate data could not be avoided by deleting a maximum of 5% of artefacts (maximum of three artefacts in a row), the time range was discarded (Mohr, Langbein & Nürnberg, 2002; Langbein, Nürnberg & Manteuffel, 2004). However, likely due to the high activity in the stress tests and the presumably bad electrode contact, data quality was too low to use any parameters of heart rate variability. In the statistical analysis, we therefore only included the baseline-subtracted heart rate, which is the average of the 20-s time intervals from the recordings of the stress test minus the averaged baseline values.

Statistical analysis

We performed all statistical analyses in R v4.0.3 (R Core Team, 2020). Seven dwarf goats and one dairy goat were excluded from administration of the treatments because they did not participate in the tests (COG) or did not take the reward (POS), i.e. showed high frequency of vocalizations, jumping at the wall or similar. Additionally, in each of the four stress tests, several goats had to be excluded from the analysis because of missing values due to technical failure of the video camera, the microphone, or the ECG device (NA: n = 13 goats, NO: n = 11 goats, NH: n = 11 goats, WH: n = 6 goats).

For each test, we performed a principal component analysis (PCA) to reduce the large set of behavioural and cardiac parameters to a smaller set of components. To improve normality before the PCA, all parameters were transformed applying Yeo–Johnson transformation with the R package bestNormalize (Peterson & Cavanaugh, 2019). We used the Kaiser-Meyer-Olkin measure of sampling adequacy (MSA) and the Bartlett test to assess the adequacy of our data for PCA (functions KMO and cortest.bartlett from psych R package; Revelle, 2021). According to Budaev (2010) an overall MSA of less than 0.5 would be unacceptable. If the overall MSA was below 0.5, the MSA of each contributing variable was calculated and variables with MSA <0.4 were dropped before the test was re-run. This procedure was continued until the overall MSA was ≥0.5. The Bartlett test was finally applied to test whether the correlation matrix is factorable (i.e., the correlations differ from zero). This procedure resulted in nine behavioural measures and one cardiac measure to be used in the different PCAs (five measures used for the NA, six for the NO, six for the NH, four for the WH; see Table 2).

The PCA was conducted with the R function principal (psych R package) using varimax rotation. To choose the final number of extracted principal components to retain, we applied nScree analysis with the plotnScree function from the R package nFactors (Raîche et al., 2013), which uses four methods: the optimal coordinates, the acceleration factor, the parallel analysis, and the Kaiser–Guttman rule. We decided to retain two components in the NA, NH, and NO for the final PCA calculation because three of the previously mentioned methods suggested so. For the WH, only the first component was retained. Because on the second component only the behavioural measure entering score loaded considerably (>|0.5|), we directly used the Yeo–Johnson-transformed entering score and not the rotated component from PCA for further analysis. To analyse the effects of the treatments (COG, POS, ISO) on the rotated PCA component scores and on the entering score from the WH, we employed linear mixed-effects models using the lmer function from the R package lme4 (Bates et al., 2015). For all models, the formula in lme4 syntax was the following:

response ~ 0 + Treatment:SelectionLine + (1|Site/Pen)

We considered a treatment effect individually for each selection line through a corresponding interaction term as fixed effect [0 + Treatment:SelectionLine]. Besides this fixed effect, a random intercept for pen nested within site (1|Site/Pen) was included to account for potential effects of the affiliation to the home pen (A–F, U–Z) and site (ET, DU). To investigate differences in stress responses between the treatments and between the selection lines, we tested contrasts for the fixed effect using the glht function from R package multcomp (Hothorn, Bretz & Westfall, 2008). P-values for fixed effect estimates and for the contrasts were obtained using Wald Z-tests (summary.ghlt function, multcomp package). Confidence bands for fixed effect estimates were obtained using the predict.MerMod function (lme4 package) in conjunction with the bootMer function (lme4) for parametric bootstrapping (104 bootstraps). Only the uncertainty in the fixed effects was taken into account (parameter re.form =  ~ 0 in predict.MerMod).

Results

Principal component analyses

Table 3 and Fig. 3 present the results of the four PCAs of the stress tests. In the PCA of the NA, we retained two rotated components (RCs, Overall MSA = 0.52). The first RC (NA_RC1) explained 30% of the total variance. It contained a high positive loading (>0.5) for the frequency of segment changes and a high negative loading (below −0.5) for the duration being inactive. Goats that loaded highly on this component were therefore labelled ‘active in NA’ (Table 2). The second RC (NA_RC2) explained 27% of the total variance. It contained high positive loadings for heart rate, frequency of vocalisations, and duration in inner segments. Because these behaviours are indicative of a high responsiveness to isolation, goats that loaded highly on this component were termed ‘reactive in NA’. In the PCA of the NO (Overall MSA = 0.56), the first RC (NO_RC1) explained 40% of the total variance. It contained a high negative loading for the duration being inactive and a high positive loading for the frequency of segment changes. Goats loading highly on this component were thus termed ‘active in NO’. The second RC (NO_RC2) explained 31% of the total variance and contained high loadings for the frequency of object contacts and for the heart rate. Because the frequency of object contacts is indicative of exploratory-like behaviours, goats that loaded highly on this component were termed ‘exploratory in NO’. In the PCA of the NH (Overall MSA = 0.57), the first RC (NH_RC1) explained 34% of the total variance. It contained high positive loadings for frequency of vocalisations and for frequency of human contacts and a high negative loading for heart rate. Goats that loaded highly on this component were therefore termed ‘sociable in NH’. The second RC in the NH (NH_RC2) explained 33% of the variation in the data. We negated its loadings (multiplication by −1) to ease interpretation. The negated component contained a high negative loading for duration being inactive and a high positive loading for frequency of segment changes. Goats that loaded highly on this (negated) component were termed ‘active in NH’. In the PCA of the WH (Overall MSA = 0.59), the single PC retained described behaviour indicative of responsiveness towards handling and weighing (WH_PC) and explained 51% of the total variance. It contained high positive loadings for the exiting score and the heart rate and a high negative loading for the weighing score. Goats loading highly on this component were termed ‘reactive in WH’.

Table 3 Principal component analysis results of each of the four stress tests (NA, NO, NH, WH) with eigenvalues, percentage of the total variance, and loadings of the rotated components (RC1 and RC2), along with communalities (= proportion of variance in the variable explained by the components).

Novel arena test (NA)	‘active in NA’ (RC1)	‘reactive in NA’ (RC2)	Communalities	
Baseline-subtracted heart rate	0.0	0.7	0.5	
Duration being inactive	−0.8	−0.1	0.6	
Frequency of vocalisations	0.3	0.6	0.5	
Frequency of segment changes	0.8	−0.1	0.7	
Duration in inner segments	−0.3	0.7	0.6	
Eigenvalue	1.5	1.3		
% of variance	30.0	27.0		
Novel object test (NO)	‘active in NO’ (RC1)	‘exploratory in NO’ (RC2)		
Baseline-subtracted heart rate	−0.04	0.9	0.8	
Duration being inactive	−0.8	−0.3	0.8	
Frequency of object contacts	0.3	0.7	0.5	
Frequency of segment changes	0.9	−0.01	0.8	
Eigenvalue	1.6	1.3		
% of variance	40.0	31.0		
Novel human test (NH)	‘sociable in NH’ (RC1)	‘active in NH’ (−RC2*)		
Baseline-subtracted heart rate	−0.6	−0.2	0.5	
Duration being inactive	0.0	−0.9	0.8	
Frequency of vocalisations	0.7	0.2	0.5	
Frequency of human contacts	0.8	−0.2	0.7	
Frequency of segment changes	0.2	0.9	0.8	
Eigenvalue	1.7	−1.7		
% of variance	34.0	−33.0		
Weighing test (WH)	‘reactive in WH’ (PC)			
Baseline-subtracted heart rate	0.6	–	0.4	
Weighing score	−0.8	–	0.6	
Exiting score	0.8	–	0.6	
Eigenvalue	1.5	–	–	
% of variance	51.0	–	–	
Notes:

* Loadings for this component were negated (multiplied by −1) for ease of interpretation.

Loadings above 0.5 and below −0.5 are shown in bold.

Figure 3 Biplots of the principal component analysis for each of the four stress tests: novel arena test (NA), novel object test (NO), novel human test (NH), and weighing test (WH).

Linear mixed-effects models

We analysed the effects of treatment and selection line on the above-described PCA components and the entering score from the WH. Fixed effect estimates are shown in Fig. 4, and fixed effect contrasts with respect to treatment and selection line are listed in Table 4 and Table 5, respectively. In the NA, we found no indication for treatment differences in activity (NA_RC1; Table 4). With weak statistical support, POS dwarf goats were more reactive than ISO dwarf goats (NA_RC2, p = 0.08). When comparing selection lines, we found that dwarf goats were less reactive to isolation than dairy goats (all p ≤ 0.03; Table 5).

Figure 4 Treatment (COG, POS, ISO) fixed effect estimates with confidence intervals from linear mixed-effects models with component scores from principal component analyses as responses.

Distributions of component scores (grey dots) are summarised as rotated kernels.

Table 4 Summary of treatment (COG, POS, ISO) contrasts from linear mixed-effects models of all stress tests (est. = estimate, s.e. = standard error, z = z-score).

The respective results for fixed and random effects are in Tables S2–S9.

	Dwarf	Dairy	
	est.	s.e.	z	p(>|z|)	est.	s.e.	z	p(>|z|)	
‘active’ in the novel arena test (NA_RC1)	
POS vs. COG	0.43	0.35	1.21	0.23	0.01	0.31	0.03	0.98	
ISO vs. POS	−0.42	0.34	−1.24	0.22	0.36	0.30	1.22	0.22	
ISO vs. COG	0.00	0.32	0.01	0.99	0.37	0.30	1.23	0.22	
‘reactive to isolation’ in the novel arena test (NA_RC2)	
POS vs. COG	0.15	0.31	0.49	0.62	−0.01	0.27	−0.05	0.96	
ISO vs. POS	−0.52	0.30	−1.76	0.08	0.00	0.26	−0.01	0.99	
ISO vs. COG	−0.37	0.28	−1.32	0.19	−0.02	0.26	−0.06	0.95	
‘active’ in the novel object test (NO_RC1)	
POS vs. COG	0.17	0.38	0.44	0.66	−0.17	0.31	−0.56	0.58	
ISO vs. POS	−0.16	0.36	−0.46	0.64	0.34	0.30	1.10	0.27	
ISO vs. COG	0.00	0.34	0.01	0.99	0.16	0.30	0.55	0.58	
‘exploratory’ in the novel object test (NO_RC2)	
POS vs. COG	0.31	0.40	0.77	0.44	0.17	0.32	0.51	0.61	
ISO vs. POS	−0.80	0.38	−2.12	0.03	0.02	0.32	0.06	0.96	
ISO vs. COG	−0.49	0.36	−1.38	0.17	0.18	0.32	0.59	0.56	
‘sociable’ in the Novel human test (NH_RC1)	
POS vs. COG	0.41	0.27	1.53	0.13	−0.03	0.25	−0.11	0.92	
ISO vs. POS	−0.44	0.26	−1.70	0.09	0.15	0.25	0.60	0.55	
ISO vs. COG	−0.02	0.26	−0.08	0.93	0.12	0.24	0.52	0.61	
‘active’ in the novel human test (NH_−RC2)	
POS vs. COG	0.31	0.31	1.00	0.32	−0.38	0.29	−1.29	0.20	
ISO vs. POS	−0.04	0.30	−0.15	0.88	0.10	0.29	0.36	0.72	
ISO vs. COG	0.27	0.30	0.90	0.37	−0.27	0.28	−0.99	0.32	
‘reactive towards handling’ in the weighing test (WH_PC)	
POS vs. COG	0.81	0.31	2.62	<0.01	0.48	0.28	1.73	0.08	
ISO vs. POS	−0.76	0.31	−2.48	0.013	0.03	0.27	0.12	0.91	
ISO vs. COG	0.06	0.29	0.20	0.85	0.52	0.27	1.91	0.06	
entering score* in the weighing test	
POS vs. COG	−0.12	0.39	−0.30	0.77	−0.17	0.35	−0.47	0.64	
ISO vs. POS	−0.14	0.38	−0.37	0.71	−0.12	0.35	−0.33	0.74	
ISO vs. COG	−0.26	0.36	−0.72	0.47	−0.28	0.34	−0.82	0.41	
Note:

* Yeo–Johnson-transformed variable (no component of the principal component analysis).

Table 5 Summary of selection line contrasts from linear mixed-effects models of all stress tests (est. = estimate, s.e. = standard error, z = z-score).

The respective results for fixed and random effects are in Tables S2–S9.

	COG	POS	ISO	
	est.	s.e.	z	p(>|z|)	est.	s.e.	z	p(>|z|)	est.	s.e.	z	p(>|z|)	
‘active’ in the novel arena test (NA_RC1)	
Dairy vs. Dwarf	−0.13	0.39	−0.32	0.75	−0.55	0.39	−1.38	0.17	0.24	0.37	0.65	0.52	
‘reactive to isolation’ in the novel arena test (NA_RC2)	
Dairy vs. Dwarf	1.03	0.39	2.67	<0.01	0.87	0.39	2.24	0.03	1.38	0.37	3.72	<0.001	
‘active’ in the novel object test (NO_RC1)	
Dairy vs. Dwarf	−0.74	0.36	−2.08	0.04	−1.08	0.38	−2.85	<0.01	−0.58	0.33	−1.77	0.08	
‘exploratory’ in the novel object test (NO_RC2)	
Dairy vs. Dwarf	0.26	0.35	0.73	0.46	0.12	0.38	0.31	0.75	0.93	0.32	2.91	<0.01	
‘sociable towards a novel human’ in the novel human test (NH_RC1)	
Dairy vs. Dwarf	0.63	0.27	2.33	0.02	0.19	0.28	0.69	0.49	0.78	0.25	3.05	<0.01	
‘active’ in the novel human test (NH_−RC2)	
Dairy vs. Dwarf	−0.65	0.31	−2.11	0.03	−1.34	0.32	−4.22	<0.001	−1.20	0.29	−4.13	<0.001	
‘reactive towards handling’ in the weighing test (WH_PC)	
Dairy vs. Dwarf	−0.44	0.32	−1.37	0.17	−0.77	0.34	−2.25	0.02	0.02	0.32	0.05	0.96	
entering score* in the weighing test	
Dairy vs. Dwarf	−0.30	0.36	−0.83	0.41	−0.35	0.38	−0.90	0.37	−0.32	0.35	−0.92	0.36	
Note:

* Yeo–Johnson-transformed variable (no component of the principal component analysis).

In the NO, we found no indication for treatment differences in activity (NO_RC1; Table 4). Dwarf goats were generally more active than dairy goats, but with varying statistical certainty within the different treatment groups (all p ≤ 0.08; Table 5). Dwarf goats from the POS treatment were more exploratory (NO_RC2) than dwarf goats from the ISO treatment (NO_RC2, p = 0.03). When comparing between selection lines, we found that ISO dwarf goats were less exploratory than ISO dairy goats (p < 0.01; Table 5).

In the NH, we found, with limited statistical support, that POS dwarf goats were more sociable than ISO dwarf goats (NH_RC1, p = 0.09). Sociability of selection lines differed within the COG and ISO treatment, with COG and ISO dwarf goats being less sociable towards a novel human than COG and ISO dairy goats, respectively (both p ≤ 0.02; Table 5). No indication for treatment differences were found for activity (NH_−RC2). Dwarf goats were generally more active than dairy goats (all p ≤ 0.03; Table 5).

In the WH, COG goats were less reactive towards handling (WH_PC) than POS goats with varying statistical certainty for dwarf (p < 0.01) and dairy (p = 0.08) goats. POS dwarf goats were furthermore more reactive towards handing than ISO dwarf goats (p = 0.013). Comparing selection lines, we found for the POS treatment that dwarf goats were more reactive towards handling than dairy goats (p = 0.02). This was not the case within the COG treatment and the ISO treatment. The linear mixed-effects model with entering score (‘entering in WH’) as response variable did not indicate differences between treatments or selection lines (Tables 4 and 5).

Discussion

We hypothesised that if cognitive testing per se has a positive impact on behaviour and cardiac activity in subsequent stress tests, goats with long-term experience with cognitive tests (COG) will show fewer responses indicative of stress than POS goats, which were not exposed to cognitive tests but experienced a similar amount of reward-associated human–animal interaction. Furthermore, we hypothesised that if reward-associated human–animal interaction has a positive impact on behaviour and cardiac activity in subsequent stress tests, POS goats will show fewer responses indicative of stress than ISO goats, which neither participated in cognitive tests nor experienced reward-associated human–animal interaction. Overall, our results from the four stress tests do not support the two hypotheses.

Effect of cognitive testing (COG vs. POS)

We had developed the hypothesis of COG versus POS based on studies which found that goats and pigs which received cognitive stimulation displayed more exploration, less fearful behaviour, and less activity in an NA and NO than animals that did not receive this cognitive enrichment (Puppe et al., 2007; Zebunke, Puppe & Langbein, 2013; Oesterwind et al., 2016). Except for the responsiveness to weighing in the WH, we did not find support for effects of cognitive testing per se (COG–POS contrast) on the responses in the four stress tests. One possible explanation for the general lack of a treatment effect is that all goats experienced a high degree of environmental stimulation already in their home pens (e.g., climbing and hiding opportunities, ad libitum hay and straw). Various studies on farm animals demonstrated that environmental enrichment, such as straw or climbing racks, can make animals less fearful and more exploratory towards an unknown object (Beattie et al., 2000; Hillmann et al., 2003; Oesterwind et al., 2016). Furthermore, all goats were confronted with additional environmental stimuli stimulation in the process of the experiment. They were repeatedly taken out of their pens and exposed to the test environment (which included novel visual, acoustic, and olfactory cues). They were also regularly handled to be equipped with loggers to measure activity in the home pen (unpublished data) and with harnesses to measure cardiac activity. In other studies, intermittent exposure to mildly stressful situations early in life, i.e. intermittent separation from the group and exposure to unfamiliar conspecifics, was found to reduce responsiveness and to improve resistance to subsequent stressors in mice and monkeys (Parker et al., 2004; Brockhurst et al., 2015). A study on horses reported that animals which spent more time outside of their stalls and were used to be ridden by two or more riders had a less pronounced adrenal response than box-stalled horses or horses with only one rider (Sauer et al., 2019). Taken together, the goats in our study may have been exposed to such a high level of intermittent stressful situations via environmental stimulation that the additionally administered cognitive stimulation through the COG treatment was rendered negligible.

Another explanation for not finding an effect of the COG treatment could be that the cognitive testing per se was not perceived as enriching by the goats. It is even possible that certain aspects of the testing procedure (i.e. uncontrollability or isolation) were experienced as negative and thus did not lead to the expected decrease in stress responsiveness in the COG goats compared with the POS goats. For example, unpredictable training events (Doyle et al., 2011; Galhardo, Vital & Oliveira, 2011) or the initial frustration caused by failure in a novel task at early stages of learning may cause stress levels to increase (Langbein, Nürnberg & Manteuffel, 2004). Particularly for social animals such as goats, isolation is a major stressor (Aschwanden et al., 2008; Siebert et al., 2011; Patt et al., 2013). Although we made sure all goats were sufficiently habituated to isolation and the COG goats willingly participated in the tests, we cannot exclude that testing was still perceived as stressful by some individuals. Also, studies that found cognitive enrichment effects on stress responsiveness had the cognitive enrichment device incorporated in the home pen, and the animals actively decided to interact with the device (Langbein, Nürnberg & Manteuffel, 2004; Puppe et al., 2007; Manteuffel, Langbein & Puppe, 2009). Such voluntary interaction with and exploration of an enrichment device may allow the animal to experience more agency over its environment, which is not only self-rewarding but also enhances the animal’s competence to deal with future challenges (Spinka & Wemelsfelder, 2011). In the current study, the experimenters but not the individual goat decided when the cognitive tests were administered. This may have led to a decreased perception of agency and could explain why we did not find reduced stress responsiveness in COG versus POS animals. The discrepancy between the findings of our study with the administered cognitive stimulation and the studies using enrichment devices therefore suggests that further research is needed to identify the positive aspects associated with cognitive enrichment (and the conditions under which it takes place) relevant for the reduction of stress responsiveness.

Effect of reward-associated human–animal interaction (POS vs. ISO)

We also hypothesised that, if goats are sustainably positively affected by the reward-associated interaction with the experimenter, POS goats would be less stress reactive than ISO goats. Overall, we did not find consistent support for this hypothesis. Only in dwarf goats and with varying statistical certainty we found some differences in stress test responses between the POS and the ISO treatment group. Dwarf goats from the POS versus those from the ISO treatment were more sociable (NH) and more exploratory (NO), presumably both indicating reduced stress and therefore supporting the hypothesis above. However, POS versus ISO dwarf goats were also more reactive towards isolation (NA) as well as more reactive towards weighing (WH). Assuming high levels in the latter two are indicative of stress, these observations would contradict the hypothesis that reward-associated human–animal interaction reduces stress responsiveness. However, the assignment of the PCA components and their underlying measures to higher or lower stress is ambiguous. For example, a higher heart rate was associated with higher responsiveness toward isolation (indicative of more stress) and with increased exploration (indicative of less stress; Fig. 3). Also, responsiveness in the WH consisted of a high negative loading for the weighing score and a high positive loading for the exiting score. This would mean that goats that were more stressed during weighing tended to be less stressed during exiting. The stress scores we adapted from D’Eath et al. (2009) might not be able to adequately capture stress responsiveness in goats depending on the type of their reactions. Whereas an active stress response might result in a low exiting score and a low weighing score, a passive stress response (e.g., freezing) would result in a low exiting score and a high weighing score. Therefore, single components or stress tests must be interpreted very cautiously if—like in our case—consistent patterns are not apparent.

However, it is worth mentioning that only for dwarf goats but not for dairy goats we found support for POS versus ISO differences at least for some stress test measures. Dairy goats seemed to be generally less stressed by human presence than dwarf goats and therefore less affected by the administered human–animal interaction (POS treatment). These differences between selection lines could be explained by genetic predisposition because selection for high productivity was shown to increase sociability towards humans (e.g., Schütz & Jensen, 2001; Lindqvist & Jensen, 2008). They could also be explained by differences in early rearing. Whereas the dwarf goats were reared with their dams until they were 6 weeks old, the dairy goats were separated immediately after birth. It was previously shown that goats raised by their dams are more reluctant to get in contact with humans (Boivin & Braastad, 1996). Early human–animal interactions were found to have lasting effects on temperament and behaviour (Lyons & Price, 1987; Lyons, Price & Moberg, 1988; Lyons, 1989). For example, goat kids gently handled at 1 week of age remained closer to a human observer and vocalised less when isolated than kids first handled at 6 months of age or not handled at all (Boivin & Braastad, 1996). Therefore, ontogeny and genetics must be considered when assessing stress responsiveness and when interacting with goats.

Conclusion

Consistent across four different stress tests, we did not find evidence that long-term experience with cognitive testing per se reduces stress responsiveness in goats. Further research is needed to identify the aspects associated with cognitive enrichment relevant for the reduced stress responsiveness found in other studies. For dwarf goats but not for dairy goats, we found support for an effect of reward-associated human–animal interactions at least for some stress test measures. This finding highlights the need to consider ontogenetic and genetic variation when assessing stress responsiveness or when interacting with goats.

Supplemental Information

Supplemental Information 1 R code.

Click here for additional data file.

Supplemental Information 2 Details regarding habituation to test environment and treatments, handling and cognitive tests and statistical test results.

Click here for additional data file.

Supplemental Information 3 Table of animals used.

Click here for additional data file.

Supplemental Information 4 Raw data: Novel arena.

Click here for additional data file.

Supplemental Information 5 Raw data: Novel object.

Click here for additional data file.

Supplemental Information 6 Raw data: Novel human.

Click here for additional data file.

Supplemental Information 7 Raw data: Weighing.

Click here for additional data file.

Supplemental Information 8 ARRIVE-Guidelines.

Click here for additional data file.

We thank the staff of the Agroscope Research Station in Ettenhausen, Switzerland, and of the Research Institute for Farm Animal Biology in Dummerstorf, Germany, for assisting in this experiment and for caring for the animals. We additionally want to thank Katrin Siebert for her help with the software The Observer, Jonas Barkhau and Jana Benner for video analysis, Charlotte Hursey and Franziska Kleindienst for help with ECG analysis, and Heather Neave for support regarding PCA analysis.

Additional Information and Declarations

Competing Interests

Author Contributions

Animal Ethics

Data Availability

The authors declare that they have no competing interests.

Katrina Rosenberger performed the experiments, analyzed the data, prepared figures and/or tables, authored or reviewed drafts of the paper, and approved the final draft.

Michael Simmler analyzed the data, prepared figures and/or tables, authored or reviewed drafts of the paper, provided statistical support, and approved the final draft.

Jan Langbein conceived and designed the experiments, authored or reviewed drafts of the paper, and approved the final draft.

Christian Nawroth conceived and designed the experiments, performed the experiments, authored or reviewed drafts of the paper, and approved the final draft.

Nina Keil conceived and designed the experiments, authored or reviewed drafts of the paper, and approved the final draft.

The following information was supplied relating to ethical approvals (i.e., approving body and any reference numbers):

All animal care and experimental procedures were performed in accordance with the relevant legislative and regulatory requirements of the corresponding country and the ASAB/ABS Guidelines for the Use of Animals in Research (ASAB/ABS, 2019). All procedures involving animal handling and treatment were approved by the Cantonal Veterinary Office, Thurgau, Switzerland (Approval No. TG04/17 – 29343) and the Committee for Animal Use and Care of the Ministry of Agriculture, Environment, and Consumer Protection of the federal state of Mecklenburg-Vorpommern, Germany (Approval No. 7221.3-1.1-062/17).

The following information was supplied regarding data availability:

The data from tests and animals and the R code used to perform the analyses described are available in the Supplemental Files.

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
