# Peer review of "Responsiveness of domesticated goats towards various stressors following long-term cognitive test exposure"

_PeerJ, doi:10.7717/peerj.12893_

## Round 0.1 · original submission · Major Revisions

I was very fortunate to receive three extremely detailed and thoughtful reviews from experts. Their recommendations ranged from reject to accept. I agree with the reviewers that there is much to like about your study; in particular the large sample and the attempt to tease apart the effects of food reinforcement and human interaction from testing.

However, I am a bit perplexed about your analytic strategy of comparing the goats by group rather than testing the effects of the actual variables you manipulated; testing, food reward, isolation. If each goat received a dummy coding representing these discrete variables, you could analyze the effects of the variables themselves rather than the groups, which themselves are merely representing combinations of the actual factors. Each group confounds at least two factors - analyzing them in this way defeats the point of your careful attempts to balance and control the factors of interest and doesn't let you examine the interaction of the factors.

The reviewers have also each listed a number of very important points where clarification is needed. I agree with their comments. Thus, I am asking you to undertake a major revision.

Reviewer 1 ·

Basic reporting

This study investigated if there is a relationship between participation in multiple cognitive tasks to reactions to novelty (environment, object, human) measured at a later stage in tow goat species; dwarf and dietary domesticated goats. Furthermore, beyond solely addressing whether cognitive testing per se is associated with changes in stress responses, the authors aimed at disentangling the effect of testing from effects of confounding factors such as the human contact and the isolation during testing. This is an important topic that benefits the field of animal cognition, as cognitive testing in a variety of captive settings is increasing within the field.
The abstract and introduction fails to transmit one of the important parts of the study, the cognitive test battery and explain what that entailed. Also, it would be necessary to refer to why the chosen tasks are considered “standardized”. The idea that some goats go through a test battery and then after (some delay, that never is specified) are measured on independent traits like response to a novel arena, object and human is very interesting, given the great efforts made by the authors to generate two adequate control groups. However, in its current state the manuscript does not bring forward this point very well. There is no mention of a cognitive test battery in the abstract, the first time the reader “learns” about this is at the end of the introduction. It would be easier for the reader to comprehend the purpose of the article if it is introduced earlier and explained more thoroughly, what tasks, what cognitive capacities are they designed to measure etc. This is nicely listed in a supplementary Table, but still would benefit from some mentioning in the manuscript. Also, the term “long-term cognitive stimulation” needs to be specified and explained, as well as “standardized” object-choice tasks. It seems like the authors tested more than solely object choice tasks, such as associative and reversal learning. Please review the literature or refer to previous work to why you choose to refer to such test battery as “standardized”.

Second, the authors claim that they are testing goats’ stress reactivity in potentially stressful situations. However, there is a large body of scientific work done on novelty response tasks (similar to the ones the authors use in this study), in the animal cognition literature and animal personality literature (few examples; Stamps and Groothuis, 2010; carter et al., 2013; Šlipogor et al., 2021), such as novel object task, open-field test, response to novel human. Such tasks are mostly employed to measure cognitive traits like neophobia, neophilia, explorative behaviors and human orientation (Damerius et al. 2017) and are not referred to as “stress tests”, as for many animals the opportunity to interact with novelty provides an enrichment activity and not necessarily direct stress. Unless there are evidence and data from goats, that already show that goats respond with stress (measured with cortisol, heart rate etc) in such tasks, I do not agree that it is appropriate to refer to these types of tests as stressors since responses to novelty can have multiple different outcomes, both neophobic (avoidance, approach delays etc) and positive explorative (manipulation, physical contact etc), which might be stimulating instead of stressing. Therefore, I advise the terminology to be reflected over and re-phrased in accordance to the published literature. For example, in the abstract the authors refer to cognitive tests as stimulating, but then presume that novelty response tasks on the other hand should be stressors, this is contradictive and needs to be elaborated on. The cognitive tasks may only be stress inducing when the nature of the task requires isolation and thus the responses of the animals are to separation from conspecifics rather than due to the task itself.

Behavioral “Reactivity” sounds like a very laboratory narrowed term, maybe better terminology behavioral “response”. However, if you expect goats to differ from other animals and therefore such tests just measure stress, please describe this with adequate references in the introduction.

Grammar and language:

The manuscript gives a general “sloppy” impression with regards to grammar and language as throughout there are many sentences are too long and would benefit from being shortened and re-phrased. In general, the manuscript does not flow easily for the reader. I suggest a native speaker or additional senior scientist to edit the text.

Passive and active voice is being mixed, for example in the first paragraph of methods section, please commit to one.

Experimental design

In line 199, the authors state “after the cognitive test phase”, please specify and provide the information on HOW LONG AFTER as you claim to test long-term responses.
A novel arena test (NA) is this the same as an open field test? If not, how do they differ? Open field tests are used to measure animal exploration behavior as well as stress responses.

weighing test (WH) – this is not really a test, more a situation also part of the routine lives of these animals. The situation when the heart rate equipment was placed onto the goats would more likely reflect a novel stressful situation rather than the weighing situation.
L206: “the goats had to be moved to a different building for these tests.” It can be possible that such move is a confunding factor contributing to stress rather than the subsequent test itself. This should be elaborated in the discussion.

If I understand correctly the Novel Objects were presented within the Novel Arena, this poses a problem as one cannot rule out the reactions of the arena or novel environment to that from the animals’ response to the novel object. Normally novel object tests are presented in a familiar environment. Also, the Novel Human test, faces exactly the same problem as the human is within the novel arena. Moreover, in the novel human test, was the person someone the goats had never seen before, please clarify. Same question for the weighing test: it says that a human led the goat onto the scale. Was this by the same person for all goats? What were the relationship to that human? such factors may influence responses. Why not use video cameras instead on the sides and subsequently score behavioral responses from the videos, as the two humans can potentially have an influence on the goats’ responses during the weighing situation.

For all tests, in the text there are no mentions of the exact behavioral measurements used in the analyses. L297: you state that you used 9 behavioral measures, even if you present them in a Table, it would be of great importance to list them in the text too.

Validity of the findings

It would be nice with any findings on the relationship between the outcomes of the test battery and the “Stress tests”. Where goats that perform better on cognitive tasks less responsive during the “stress tests”? This was a question that one expected from the introduction but is not addressed in the result section at all.

L382: This would be easier to read if it is shortened and would refer to the two hypotheses - present them clearly in the end of the introduction and in result section briefly, but then no need for a long repetitive section in the discussion.

One possibility to why there was a lack of connection between the cognitive testing and the following stress tests, is the fact that with the current experimental design it is impossible to disentangle interest in objects from isolation stress. One would need to measure interest in novel object in a familiar, every day environment to ensure that such response does not intervein with the stress of being separated into a novel arena. The authors briefly “touches” this topic L445-448, but would have to elaborate on their findings more clearly before interpreting the results.

L462-470: In this section the authors nicely discuss potential explanations of why different groups of goats responded differently in their tests and acknowledge the variation in rearing backgrounds as well as upbringing experience. It would be nice to explore if these rearing influences had an effect on the outcomes of the cognitive test battery as well as the novelty response tests.

L469: phenotypic variation? but what you mean is rearing and background history.
Concluding comment may represent a line of misinterpretations: “We found that long-term experience with standardized cognitive testing per se does not reduce stress reactivity of goats in subsequent stress tests.” This is an overstatement given the lack of definitions in the terminology in the first place and given that the stress response was not properly disentangled from isolation response.

Additional comments

Detailed comments:
ABSTRACT
L23: Very long and complicated introductory sentence.
L41: This sentence is a bit out of place and not clear on how it connects to the rest that has been said in the abstract.
INTRODUCTION
L49: purposes instead of aim?
L51: one space too much
L52: I am not sure what the word learning at the first part of this sentence adds, it seems a bit unexplained and should therefore better be left out or one should explain the context of it here. I think the authors are trying to say something like: Because cognitive tests can potentially cause the participating animal stress, as provided tasks often requires successful learning by the animal before it experience positive emotions (Langbein, Nürnberg & Manteuffel, 2004). I would also take out “initial stages”, as there are also cognitive tests that demand attention and learning for a prolonged time.
L56: “execution”? maybe better completion, participation?
L57: “were”, maybe better has been?
L59: “a call-feeding station”. The reader does not know what this is, please elaborate and explain.
L60: “that the animals were” better that those individuals… or those subjects were… compared to … conspecifics.
L61: Who are they? Specify please.
L65-69: too long sentence, please split and clarify.
L67: “be isolated” from their group of conspecifics? Please specify.
L73: in which species? It would be nice if the reader was given that information.
L92: bind the paragraph to previous with some wording like: “Moreover”, “Furthermore” etc.
L96: “was” replace with “has been”
L103: “extensive husbandry conditions”, in what way? Please clarify and explain.
L110: “were” replace with “has been”
L111: “highly developed cognitive capacities” – like what capacities? Please explain rather than just make a blunt statement.
L114: “by means”, replace with “using”
L116-123: This section is rather suited for the methods part; it makes the introduction harder to grasp.
METHODS
L131-132: No need to replicate the word “Animals” in both headings. Delete “Animals” from L131.
L133: Delete “namely” and replace with simply “;”
L138: “few exceptions”. Exceptions always has to be mentioned and clarified.
L138: “aim”? replace with “purpose”
L139-140: “The potential milk yield of Nigerian dwarf goats does likely not exceed 0.3 kg/day (Akinsoyinu, Mba & Olubajo, 1977)” Unclear wording, do you mean: generally does not exceed?
L140-141: Wrong grammar, please rephrase this sentence.
L146: “and artificially raised” please explain what this means.
L183: do you mean habituation process of the goats or habituation tests? Please specify.
L189-190: The sentence does not flow, please correct grammar.
L190: “administered”? what does this complex word add? Why not simply individuals of ISO treatment
L225: This sentence makes no sense, needs elaboration.
L282: As you exclude these animals with the definition of them being too stressed out during the cognitive test battery, please give the reasons to why you believe they were stressed out.

Suggested literature (just a few)
Stamps, J., & Groothuis, T. G. (2010). The development of animal personality: relevance, concepts and perspectives. Biological Reviews, 85(2), 301-325.
Carter, A. J., Feeney, W. E., Marshall, H. H., Cowlishaw, G., & Heinsohn, R. (2013). Animal personality: what are behavioural ecologists measuring?. Biological Reviews, 88(2), 465-475.
Šlipogor, V., Massen, J. J., Schiel, N., Souto, A., & Bugnyar, T. (2021). Temporal consistency and ecological validity of personality structure in common marmosets (Callithrix jacchus): A unifying field and laboratory approach. American journal of primatology, 83(2), e23229.
Damerius, L. A., Forss, S. I., Kosonen, Z. K., Willems, E. P., Burkart, J. M., Call, J., ... & van Schaik, C. P. (2017). Orientation toward humans predicts cognitive performance in orang-utans. Scientific Reports, 7(1), 1-12.

·

Basic reporting

(Comments listed in order of importance)

• Line 343 onwards: Linear mixed-effect models section – it is more conventional to report the test statistic etc., not just the p-values
• Lines 30-33: Especially for the POS treatment and somewhat for the COG treatment I feel you should either change the treatment names or describe them in a way that it is obvious where they derive from. In respect to the POS treatment, this is better described at the end of the introduction (Line 117), but the link between the COG treatment name and description becomes more unclear (Lines 115-116). This will enhance memorability to readers as to what treatments involved further on in the text.
• The term selection lines is used throughout (e.g. lines 35, 133), but I feel this it is used inappropriately as authors were not just comparing between two distinct breeds as they used three different dairy goat breeds and cross-breeds.
• Supplementary material: Table s2 onwards: explain the shortened terms used in the description above, i.e. est. = effect size?, s.e. = standard error etc.

Experimental design

(Comments listed in order of importance)

• How long did it take for the goats to clear these treatment trials and what was the average duration between completion of the COG condition and the start of the experimental trials? This is important information and should be mentioned in the methodology section, but I cannot find this written anywhere
• What was the average time interval between the stress-tests?
• Was it always the same person conducting the battery of tests for each goat? This could at least have important implications for the novel human test
• As human associated rewarding is a major factor in the experimental design, I feel what other forms of human contact did the goats receive before onset of tests (e.g. husbandry routines etc.) should be explained in the methodology section
• Line 171 onwards: Treatment groups section: it would be very useful for readers to have the tests briefly described in a bit more detail to better understand them without having to be referred to another document. This supplementary document would be useful to keep in, but I would look for a bit more detail of what tests are involved and how regularly they were exposed to testing
• Supplementary material: Did any goats fail to reach the success criteria for these cognitive tests? If so, what happened to such goats?
• Supplementary Materials: Table S1 – make the description of the visual discrimination task clear, for example by explaining only one of the cups was baited
• Line 198: did you consider order effects/ randomisation of test conditions? Why did you choose to do the tests in a standardised order?
. Supplementary materials: it would be useful to have a photograph or diagram of each test paradigm used
• The authors used three breeds of dairy goat and some mixed breeds. Did you look for the effect of breed on results, rather than just splitting comparisons between dairy breeds vs pygmy goats?
• The authors mentioned using an external microphone to look at acoustic responses. Did you investigate the effect of stress tests on acoustic parameters relative to treatment condition?

Validity of the findings

Comprehensive information has been provided regarding subjects, the raw data and even R scripts used in analysis.

However, in order to make the behavioural data set more robust, I was wondering whether you had considered using a secondary coder other than just for the weighing test?

Additional comments

N/A

Reviewer 3 ·

Basic reporting

This study how long-term experience with cognitive test per se does not reduce stress reactivity of goats in subsequent stress tests. I believe that this study is essential in order to absolutely validate any results obtained in experiments carried out with domestic species (cognition, behaviour, etc). On the other hand, the study provides relevant information that needs to be further studied to ensure proper management of livestock.
The language as well and the English are at highest levels. The manuscripts flow easily its informative and contains all the background information needed. All data, tables, figures and raw data included in the supplementary material are satisfactory well ordered and informative. The objective of the study is sound and the hypotheses too.
The information contained in lines 100-108 regarding the two breeds of goats perhaps could be changed to the discussion, since its mainly the justification of using them and why is important. Nevertheless, if the paragraph maintains its current location the quality of the manuscript remains intact.
Overall, I find the manuscript very pleasant to read, informative and comprehensive and I have no punctual comments.

Experimental design

I believe that the topic of this manuscript is highly relevant, that it is original and fall in the scope of the journal. The hypotheses are clear and sound. I believe that the experimental design is indeed answering all the research questions formulated by the authors. In my opinion, the experimental design is complex, however, the authors found a way to address all the questions and answer them including all possibilities. The methods are well described and easily reproducible.

I believe that this study addresses a very important subject which is usually just “solved” by a period of “habituation” before running cognitive or behavioural tests. However, it is imperative to know if the long-term cognitive test exposure per se has a positive impact, backing it up with physiological (cardiac frequency) and behavioural assessment.

The experiments are clear the justification of the different groups investigated is well explained.
I have no punctual comments about the experimental design or the methods used due to the clarity of the manuscript.

Validity of the findings

In my opinion, the results of the study are absolutely valid, mainly for the good and neat experimental design. Many researchers still reject studies with negative results. I find these negative results very important and necessary in order to design further research. The fact that cognitive testing per se, usually believed to decrease the stress in subjects exposed to further tests, and that the reward-associated human-animal interaction having a positive impact in the subject’s stress, are not supported by the results obtained is very imperative. I believe that the findings of this study highlight the big gap of knowledge that is still missing and the clear necessity that more researching study cognitive testing, habituation, human-animal interaction is greatly needed. Additionally, it points out how other studies might obtain results that lacks validity.


The results are very well discussed, the statistical analysis well done and the authors have provided alternative justification of the results obtained. The study highlights how important is to keep investigating these subjects that are essential for all researchers studying, cognition, behaviour and animal welfare.

I have no punctual comments since the I find the discussion and conclusion of the results absolutely valid. I must add that the manuscript is interesting and very pleasant to read.

Additional comments

Dear authors

I found your manuscript interesting and very pleasant to read. Overall, it is clear, the background information provided covers well the topic and the experimental design is absolutely sound. Your data figures and statistical analyses are satisfactory and, in my opinion, validate your findings. I would encourage you to keep designing studies to learn more about the stress experienced by livestock in general. The use of two breeds is also highly commendable and increase the validity of your results. The discussion is compelling and the possible explanations you provide are sound and well backed up with great references. I honestly did not find any punctual comments to make, the manuscript is excellent as it is.

---

## Round 0.2 · Minor Revisions

Thank you for your careful and detailed response to the last round of reviews. Two of the previous three reviews looked at the revision and are satisfied that the manuscript can now be published. One reviewer has some minor comments that if addressed will improve the clarity of the manuscript. I have a few minor edits myself. Once these final, very minor edits are made, I’ll be happy to formally accept the manuscript.

- Please provide information about the number of goats tested in the abstract.
- Please insert the species name at first mention.
- Delete the extra . on line 84 of the tracked change document.
- Please enclose all punctuation within quotations throughout.
- Please insert ',' after e.g. and et al.
- Please change explorative to exploratory or something similar.

Reviewer 1 ·

Basic reporting

I believe the authors have made necessary efforts to improve the points raised during first round of revisions.

Experimental design

No additional comments. I am satisfied with the clarifications made in the new version and all my previous raised issues were properly addressed.

Validity of the findings

Sufficient explanations have been added to improve the manuscript. However, as the authors state "the aim of the current study was to assess the impact of cognitive testing on subsequent performance in conceptually different cognitive test". I would advice the authors to be careful using the word cognitive testing if what they refer to is the perception of novel humans/ objects during stress.

Additional comments

No additional comments.

·

Basic reporting

Although the language used is largely very good, there are a few suggestions I would like to make:

L32: ‘in ‘the’ form of visual …
L38: You should make clear animals were isolated from conspecifics as they were not completely isolated given the presence of experimenters
L71: First = Initial? And also has = have
L82: those= these
L83, L541-542, L565, L565, L669: the term explorative although technically correct, is little used, and I would consider using exploratory instead
L157: 'they are usually kept in ‘petting zoos’ or by hobby breeders in Europe'
L234: ‘segment changes:’ this terminology is made clear after reading the methods section, but given it is mentioned in the introduction, readers at this stage will not understand what this means
L280: ‘except of eight animals’ = with the exception of eight animals
L345-L366: It needs to be made a bit more clear that the order of which the tests are explained is the order in which they took place
L459: would be best to add the test you are using, i.e. Spearman’s rank rather than just relying on the shorthand rs
L675: not unambiguous = ambiguous

Experimental design

no comment

Validity of the findings

I would just additionally like to ask, why for the novel human and object test did you use space utilisation of the inner versus outer segments instead for example time spent in or adjacent to the square where the object/ human is positioned?

Additional comments

I would like to make clear these suggestions are only for very minor improvements, and if largely unheeded, it will not make a substantial impact to the quality of the manuscript which I feel is very high. The author has largely acted upon or replied to my previous comments and would like to thank them for doing so. I am very satisfied with their responses and revisions and feel the manuscript has improved since I last read it.

---

## Round 0.3 · accepted · Accept

Thank you for attending to these final minor points.